# Spatial-temporal characterization of cropland abandonment and its driving mechanisms in the Karst Plateau in Eastern Yunnan, China, 2001–2020

Jingyi Wang[1,2], Jiasheng Wang 📷[1,2]\*, Jianhong Xiong[1,2]\*, Mengzhu Sun[1,2], Yongchao Ma[1,2]

**1** Faculty of Geography, Yunnan Normal University, Kunming, Yunnan Province, P. R. China, **2** The Engineering Research Center of GIS Technology in Western China of Ministry of Education of China, Yunnan Normal University, Kunming, Yunnan Province, P. R. China

\* jersonwang@ynnu.edu.cn (JW); xjianhong@foxmail.com (JX)

**Data Availability Statement:** The data are all contained within the manuscript file.

## Abstract

The karst plateau is dominated by mountainous and hilly landforms, with low mechanization level of cropland, high difficulty of cultivation, and obvious phenomenon of cropland abandonment, which threatens regional food security. This study aims to analyze the spatial-temporal variation and its driving mechanisms of abandoned cropland in the Karst Plateau in Eastern Yunnan, China (KPEYC) between 2001 and 2020. To achieve this goal, 18 key factors from population, economic environment, cropland attributes, and farming conditions are selected. Moreover, correlation analysis, geodetector, and regression analysis methods are applied from three perspectives: temporal change, spatial distribution and spatial-temporal change. The results show that: (i) The cropland abandonment rate (CAR) in the KPEYC shows a fluctuating trend, with an average value of 9.78%, and the spatial distribution shows a pattern of "high in the center and low in the south and north". (ii) From the perspective of temporal change, gross value of agricultural production, and gross value of industrial production have the largest correlation coefficients with CAR. (iii) The explanatory power of gross tertiary industrial production, gross value of industrial production, followed by soil thickness. (iv) Gross value of agricultural production, and gross tertiary industrial production are the core driving forces for the spatial-temporal change of CAR. The higher the gross value of agricultural production and gross tertiary industrial production, the lower the CAR. elevation, soil thickness, and traffic mileage are the main driving factors for the spatial-temporal change of CAR. The study indicates that economic factors are decisive for cropland abandonment in the KPEYC. Based on the results, this study can provide decision-making support for local prevention and control of cropland abandonment, and the local community needs to promote land transfer and concentration and local urbanization according to local conditions, improve agricultural policies, improve farming conditions, etc. in order to increase farmers' enthusiasm for production, promote the rational use of cropland, and solidly push forward ecological restoration and management, optimize ecological spatial

**Funding:** The National Natural Science Foundation of China (Grant No. 41961056) was used to purchase data for this study; The Yunnan Province high-level Talent Training Support Program (Grant No. YNWR-QNBJ-2020-053) was used for outgoing research; Yunnan Normal University 2023 Graduate Student Research and Innovation Fund (Grant No. YJSJJ23-B149) was used for experimental processing.

**Competing interests:** The authors declare that they have no known competing financial interests or personal relationships that could have appeared to influence the work reported in this paper.

patterns, manage serious areas of rocky desertification, and appropriately alleviate the contradiction between people and land.

## 1 Introduction

Abandoned cropland usually refers to the phenomenon of cultivated land no longer cultivated, resulting in barren land in the form of grasslands, shrublands, woodlands, and other types of land [1, 2]. In the 21st century, China's industrialization and rural urbanization process is constantly advancing, and a large number of rural workers are moving to the cities, resulting in a shortage of agricultural workers [3, 4]. The phenomenon of cropland abandonment is becoming increasingly prominent, leading to the waste of cropland resources, the reduction of local crop yields, the inability to achieve self-sufficiency in food production, the reduction of farmers' income, the hindrance of agricultural development, and the threat to regional food security. Studying the driving factors of cropland abandonment and revealing its driving mechanism is important for protecting designated cropland, preventing further abandonment, increasing land productivity, and facilitating rural development and revitalization in China [2].

Cropland abandonment is the result of multiple factors at different scales [5–7]. Scholars at home and abroad have conducted extensive research on the driving mechanisms and influencing factors, mainly analyzing the factors of abandonment from the perspective of the social system and land use change [8–11]. Because of the differences in natural and social environments, the driving mechanisms of cropland abandonment vary widely across countries. For example, Prishchepov et al. argued that the collapse of the Soviet Union led to changes in the social system that increased abandoned cropland [12]. On the other hand, in developed countries such as Europe and Japan, urbanization and industrialization are the main drivers of cropland abandonment. Studies have also suggested that the Common Agricultural Policy and the policy of subsidizing fallow land are important factors contributing to cropland abandonment in Europe [3, 13]. Research on cropland abandonment in China primarily focuses on regions such as the Loess Plateau, the low hills of North China, and the Taihang Mountainous Region. Scholars have mainly studied the causes of abandonment and countermeasures from the socioeconomic perspective [3, 4]. For example, in Henan Province, the outflow of population, low agricultural production yield, and fragmented cropland have resulted in a serious phenomenon of cropland abandonment [11, 14]. Similarly, the terraced fields in mountainous areas of China are affected by low mechanization, irrigation conditions, limited transportation, and labor force, leading to large-scale abandonment [3].

Commonly used methods for analyzing driving mechanisms include sample survey analysis, meta-analysis, principal component analysis, correlation analysis, regression analysis, and geodetector analysis. The survey method is suitable for discovering the driving factors of geographical phenomena. For example, Shijie Dong investigated the degree of abandonment of terraced rice fields in China through national sampling and further analyzed the spatial differentiation characteristics of the abandonment of terraced rice fields and its driving factors [3]. Correlation analysis quantifies the strength of the relationship between two sets of multivariate random variables, revealing the inherent interaction between two sets of variables [15], Xingzhu Yang used correlation analysis and other methods to explore the differences in the characteristics of the quality of the rural habitat in the region [16]. The regression analysis method is a mathematical and statistical method used to examine the correlation between variables, investigate the correlation between variables to find out the intrinsic law, and then

predict and estimate the trend of variables [17]. The geodetector analysis is a set of statistical methods used to detect spatial heterogeneity and reveal its driving force, which can evaluate the correlation between factors in terms of spatial distribution [18]. Dingyang Zhou used geodetector analysis to reveal the formation mechanism of cropland abandonment in Henan Province [11]. From the perspective of research methods, the existing research on cropland abandonment lacks an analysis of the driving mechanism of spatial and temporal changes. From the perspective of driving factors, existing studies mainly analyze the driving mechanisms of cropland abandonment in terms of parcels or households [19–21]. Few studies have been conducted to analyze the driving factors of cropland abandonment in terms of time scales, starting from the basic unit of the county. The combination of temporal and spatial studies of cropland abandonment can reveal more thoroughly reveal the driving mechanism of the phenomenon.

In summary, this study introduced the research progress, influencing factors and methods of cropland abandonment, but karst landscapes cover about 20% of the land surface, but there are very few studies on cropland abandonment in this particular landscape. The Karst Plateau in Eastern Yunnan, China (KPEYC) is one of the largest areas in China known for its karst landscape distribution. Cropland in the region is fragmented, the ecological environment is fragile, per capita cropland is small, the quality of cropland is generally not high, and there are insufficient reserve resources of cropland, so it is difficult to rely solely on agricultural incomes to support household expenditures, and pronounced human-land conflicts. As a result, many rural workers migrate to cities and towns for work, resulting in significant cropland abandonment. This study aims to address three aspects: (i) the spatiotemporal distribution characteristics of abandoned cropland; (ii) the main driving factors of cropland abandonment; (iii) the driving mechanisms of cropland abandonment in the eastern part of Yunnan Province.

## 2 Study area and data

### 2.1 Study area

KPEYC is a region located in the eastern part of Yunnan Province. It includes Qujing City, Zhaotong City, Wenshan Zhuang, and Miao autonomous prefectures. It is bordered by Sichuan Province to the north, Guangxi Autonomous Region and Guizhou Province to the east, and Kunming City and Honghe Hani autonomous prefecture to the west (Fig 1). The study area covers about 84,100 square kilometers, with an average elevation of about 2,000 meters. The terrain varies from high in the northwest to low in the southeast, and the region is characterized by the intersection of mountains and rivers and complicated topography. The geological structure of the area is complex, with carbonate rocks widely distributed and of large size, making it a typical karst geomorphologic area. The plateau monsoon climate of the subtropical and temperate regions is dominated by four seasons, with small temperature differences between them but large differences between regions. Long daylight hours, abundant rainfall, and distinct rainy and dry seasons exist.

KPEYC is the most important grain production area in Yunnan Province, The cropland area in KPEYC accounts for 35% of the province's cropland. With major crops including corn, potatoes, legumes, rice, and wheat. Among them, corn is the most widely grown and distributed crop. Cash crops mainly include oilseeds and tobacco. In 2022, the resident population of the East Yunnan region is 14.158 million people, with a cultivated area of 27.7169 million hectares. The per capita cropland is 1.9 hectares per person, and the GDP in 2021 is 543.321 billion yuan. The terrain in the region is hilly, and the cropland is fragmented. The level of agricultural mechanization is low, and the region is affected by rocky desertification, resulting in

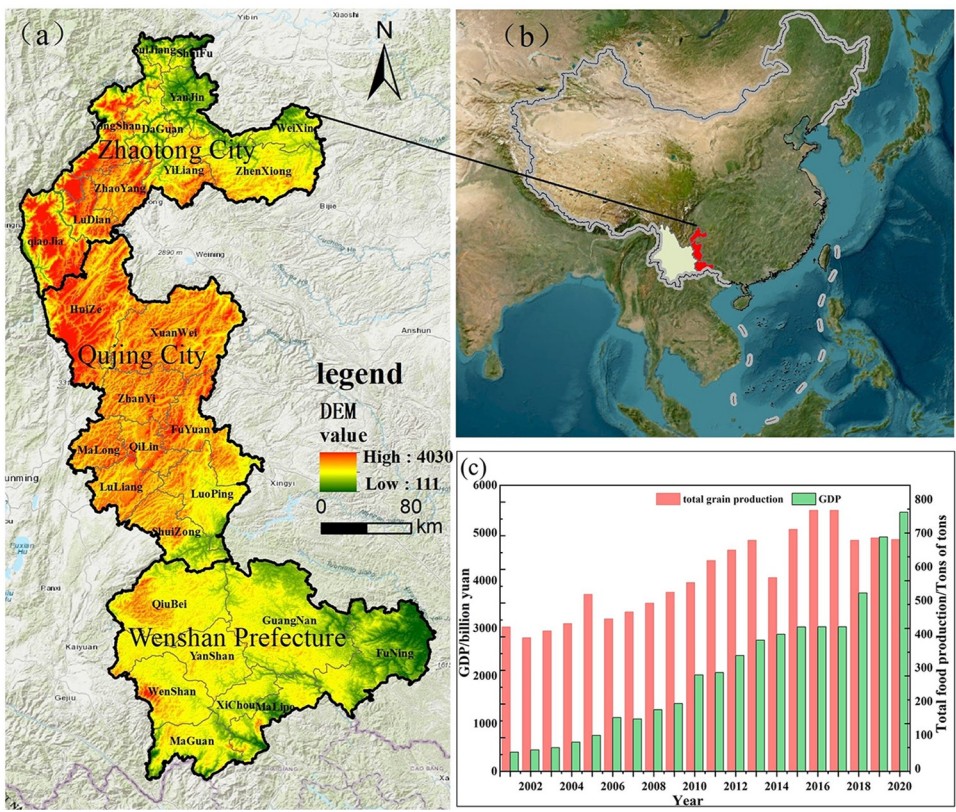

**Fig 1. The location of the study area.** (a) The topographic map. (b) The KPEYC in China. (c) The 20-year change of total grain output and GDP in KPEYC.

serve soil erosion. The ecological environment is fragile, and the destruction of cropland is severe. The conflict between humans and land is pronounced.

## 2.2 Data and pre-processing

**2.2.1 The abandoned cropland dataset.** The dataset of abandoned cropland in KPEYC from 2001 to 2020 was obtained from Zhao's [22] research. This dataset had a spatial resolution of 30m and an accuracy of 0.94. Using this data, the area of abandoned cropland in each county in eastern Yunnan was calculated by analyzing the KPEYC county layer and dividing it by the cropland area of the previous year. Thus, the cropland abandonment rate (CAR) for each county for each year between 2001 and 2020. The formula is as follows:

$$AR_i = \frac{AA_i}{CA_{(i-1)}} \times 100\% \tag{1}$$

where $AR_i$ is the CAR in the year $i$, $AA_i$ is the area of cropland abandonment in the year $i$, and $CA_{(i-1)}$ is the area of cropland in the year before the year $i$.

**2.2.2 Natural geographic data.** The natural geographic data used included data on cropland attributes and farming conditions. These eight categories were elevation, slope, slope direction, average air temperature, average precipitation, average sunshine duration, soil thickness, and soil organic matter content. The data were based on the county as the basic unit. The elevation data were obtained from ASTER GDEM V3 of the geospatial data cloud (www.gscloud.cn/search), with a resolution of 30 m. The data used for the production of digital maps

in the study area was obtained from the Natural Earth (http://www.naturalearthdata.com/). The number of slopes and slope directions was calculated based on the elevation. In addition, the average elevation, average slope, and average slope direction of each county were calculated using GIS software. The data on average air temperature, average precipitation, and average sunshine duration were obtained from the Qujing Yearbook, Zhaotong Yearbook, and Wenshan Prefecture Yearbook from 2001 to 2020, which were downloaded from the China Knowledge website (http://www.cnki.net). Soil thickness and soil organic matter content data were obtained from the official website of the Food and Agriculture Organization of the United Nations (https://www.fao.org/) with a resolution of 1 km. The average soil thickness and soil organic matter of each county were statistically obtained using GIS software.

**2.2.3 Socio-economic data.** The socio-economic data include various indicators such as year-end total population, agricultural population, rural workers, the number of primary and secondary school students, gross value of agricultural production, gross value of industrial production, gross tertiary industrial production, per capita net income of farmers, traffic mileage, and agricultural machinery power. The data on the year-end total population, the agricultural population, the rural labor force, the number of students, and the agricultural machinery power data were taken from the China Statistical Yearbook 2001–2020, which was downloaded from the website of the National Bureau of Statistics of China (http://www.stats.gov.cn). The gross value of agricultural production, gross value of industrial production, gross tertiary industrial production, per capita net income of farmers, traffic mileage data were taken from the 2001–2020 Yunnan Provincial Statistical Yearbook, Qujing Yearbook, Zhaotong Yearbook, and Wenshan Prefecture Yearbook, which were downloaded from the China Knowledge website (http://www.cnki.net).

## 3 Methods

This study aims to investigate the driving factors of cropland abandonment, and it consists of three main parts: driver selection, driver analysis, and driver mechanism construction. In the first part, we review the existing literature and select the driving factors associated with cropland abandonment. In the second part, correlation analysis and the geodetector method are used to analyze the main factors responsible for the spatial and temporal variation of CAR. Finally, in the driving mechanism construction part, we construct a linear regression model based on data of major factors and CAR to explore the driving mechanisms of cropland abandonment.

### 3.1 Selection of potential driving factors

Cropland abandonment results from human utilization of cropland, which in turn is influenced by human decision-making. Making informed decisions requires a comprehensive consideration of both human and land factors. Human factors such as labor force and economic income play a crucial role in determining whether or not to abandon cropland, while land factors such as the attributes of farmland and farming conditions influence the location of abandonment. Given that this paper's research unit is the county, we have selected population, economic environment, cropland attributes, and farming conditions as the key influencing factors to be analyzed (Table 1).

### 3.2 Time series testing

To test the stationarity of the abandoned cropland over time series, we used the ADF (Augmented Dickey-Fuller test), a rigorous statistical testing method. It determines whether a time series is stationary by comparing the test statistic with its corresponding critical values. If the

**Table 1. Potential driving factors of cropland abandonment.**

| Level Indicators | Level 2 Indicators | Unit | Description |
|---|---|---|---|
| Population | Year-end total population X1 | *Million people* | The larger the population, the less likely it is that cropland will be left fallow [1]. |
| | Number of students in primary and secondary schools X2 | *Million people* | It can be used as supplementary labor for farming [14]. |
| | Rural workers X3 | *Million people* | The lower the amount of labor, the higher the likelihood of abandonment [5]. |
| | Agricultural population X4 | *Million people* | A decrease in the rural agricultural population makes abandonment more likely [1]. |
| Economics Environment | GDP X5 | *Billion yuan* | The greater GDP, the more likely it is to be abandoned [4]. |
| | Gross value of agricultural production X6 | *Billion yuan* | The more developed the agricultural is, the lower likely it is to be abandoned [7]. |
| | Gross tertiary industrial production X7 | *Billion yuan* | The larger the output value of tertiary industry is, the more likely it is to be abandoned [13]. |
| | Per capita net income of farmers X8 | *Ten thousand yuan* | The higher the income of farmers, the lower the likelihood of abandonment [5]. |
| Cropland Attributes | Soil thickness X9 | *Meter* | The thicker the soil layer, the less likely it is to be abandoned |
| | Soil organic matter content X10 | *Grams per kilogram* | The higher the quality of the soil, the less likely it is to be abandoned [5]. |
| | Average sunshine duration X11 | *Hour* | The better the light, the less likely it is to be abandoned. |
| | Average air temperature X12 | *Celsius* | The better the temperature, the less likely to be abandoned [9]. |
| | Average precipitation X13 | *Millimeter* | The richer the water resources, the less likely it is to be abandoned [9]. |
| Farming Conditions | Elevation X14 | *Kilometer* | The higher the altitude, the more prone to abandonment [14]. |
| | Slope X15 | *Degree* | The steeper the slope, the more likely it is to be abandoned [11]. |
| | Slope direction X16 | *Degrees* | The sunnier the slope, the better the light, the less likely it is to be abandoned [14]. |
| | Traffic mileage X17 | *Thousand Kilometers* | Poor access to cropland can lead to abandonment [11]. |
| | Agriculture machinery powerX18 | *Million Watts* | The lower the degree of mechanization, the higher the likelihood of abandonment [5]. |

test statistic is less than the critical value and the p-value is less than 0.05, we can reject the null hypothesis and conclude that the time series is stationary. Results are shown in Table 2. As can be seen from Table 2, when the difference is at order 0, the significance P-value is 0.001, which indicates significance at the given level. Thus, the null hypothesis is rejected, and the series is considered to be a stationary time series.

## 3.3 Drivers of temporal changes of CAR

They have not changed much over the years, as factors such as cropland attributes and farming conditions have remained relatively stable. In this paper, we take the entire study area as the unit of calculation and use correlation analysis to explore the degree of correlation of population, economic environment, two farming conditions factors and changes in CAR. It mainly involves the calculation of indicators and correlation coefficients.

**Table 2. ADF time series test results table.**

| variable | Difference Order | t | P | AIC | 1% | 5% | 10% |
|---|---|---|---|---|---|---|---|
| CAR | 0 | -4.132 | 0.001 | -56.957 | -3.833 | -3.031 | -2.656 |
| | 1 | -5.414 | 0.000 | -51.999 | -3.889 | -3.054 | -2.667 |
| | 2 | -0.904 | 0.787 | -50.326 | -4.332 | -3.233 | -2.749 |

(1) Indicator calculation. The average value of each indicator in each year is taken as the indicator value of the study area, and the time series of the indicator value is the independent variable $X$. The average value of the annual cropland abandonment area/abandonment rate in the region is taken as the dependent variable $Y$.

(2) Spearman correlation analysis [23]. Correlation analysis aims to determine the correlation coefficients between each variable and the dependent variable and to determine whether they are significant or not using statistical tests. Spearman's correlation coefficient is also known as the rank correlation coefficient, which requires two conditions to be met, 1) the variables contain ranked variables or the variables do not follow a normal distribution, 2) there is a monotonic relationship between the two variables. This approach has a wider range of applications than Pearson's correlation coefficient. The formula is as follows:

$$R_s = \frac{\sum_{i=1}^{N}(P_i - \bar{P})(S_i - \bar{S})}{[\sum_{i=1}^{N}(P_i - \bar{P})^2 \sum_{i=1}^{N}(S_i - \bar{S})^2]^{\frac{1}{2}}} = 1 - \frac{6\sum_{i=1}^{N} d_i^2}{N(N^2 - 1)} \tag{2}$$

In the formula, $R_s$ represents the correlation coefficient between the drivers and CAR, and the value range $R_s$ is [–1,1]. When $R_s>0$, the impact factor is positively correlated with the CAR; When $R_s<0$, the impact factor is negatively correlated with the rate of cropland abandonment. The larger the absolute value $R_s$, the greater the correlation between the impact factor and the CAR. $n$ is the capacity of the sample; $P_i$ and $S_i$ represent the average values of the observed data $i$; $\bar{P}$ and $\bar{S}$ respectively represent the average ranks of the variables $x_i$ and $y_i$; and $d_i = P_i - S_i$.

## 3.4 Drivers of spatial variation of CAR

To explore the driving factors of the spatial variation of cropland abandonment, this paper analyzed the driving factors of the spatial distribution of cropland abandonment using the geo-detector model [18]. Cropland abandonment is a complex process, and most of the existing studies of abandonment drivers have used traditional econometric models, focusing on the effects of individual factors on abandonment and paying little attention to the interactions of multiple factors. geodetector are a set of statistical methods for detecting spatial heterogeneity and revealing the driving forces behind it. Each time-varying factor was calculated as the average value of 2001–2020 by county as the factor value. The invariant factor was added as the independent variable $X$, and the average value of the CAR from 2001 to 2020 was taken as the dependent variable $Y$. Geodetector was used to detect the driving factors of the spatial distribution of the average CAR.

**(1) Spatial stratification and heterogeneity detection.** The geodetector can detect the spatial stratified heterogeneity of the variable $Y$ and the degree of explanation of a certain factor ($X$) on the spatial variability of the CAR ($Y$), which is measured by the q-value, and its calculation formula is Eq (3):

$$q = 1 - \frac{1}{n\sigma^2} \sum_{h=1}^{L} n_h \sigma_h^2 \tag{3}$$

Where L is the number of partitions of the variable $X$, $n$ is the sample size of $Y$, $\sigma^2$ is $Y$ the overall variance of $Y$, $n_h$ is the $h$ sample size of the stratified samples $h$, and $\sigma^2$ is the variance in the $h$ partitions. The $q$ of factors value range is [0,1], the larger the value, the stronger the explanatory power of the independent variable on the abandonment rate $Y$.

**Table 3. Interactive detection between different factors $X_i$.**

| Basis of judgment | Type of interaction |
|---|---|
| $q(X1 \cap X2) < \text{Min}[q(X1), q(X2)]$ | Nonlinear strengthening and weakening |
| $\text{Min}[q(X1), q(X2)] < q(X1 \cap X2) < \text{Max}[q(X1), q(X2)]$ | Single-factor nonlinear attenuation |
| $q(X1 \cap X2) > \text{Max}[q(X1), q(X2)]$ | two-factor enhancement |
| $q(X1 \cap X2) = q(X1) + q(X2)$ | independent |
| $q(X1 \cap X2) > q(X1) + q(X2)$ | nonlinear enhancement |

**(2) Interaction detection.** The geodetector also has the function of interaction detection, which can analyze the interaction between different factors, i.e., to assess whether the factors $X_1$ and $X_2$ together increase or decrease the explanatory power of the variable $Y$, and also to calculate whether the factors are independent of each other. The calculation method is to calculate the explanatory power of the two factors $q_1$ and $q_2$, and then calculate the $q$ value of the new polygon formed by the tangent of the superposition of the polygon of the two variables, and compare it with the explanatory power of a single factor, and the relationship between the two factors is shown in Table 3. Interaction detectors detect interactions between factors, determine whether variable interactions exist between indicators, and determine the strength, direction, nonlinearity, and nonlinearity of interactions. In this study, an interaction detector was used to detect the effect of a superposition of cropland abandonment indicators on the CAR.

## 3.5 Drivers mechanism of spatial-temporal variation of cropland abandonment

This paper uses the regression analysis method to do linear regression analysis method of the influencing factors and dependent variables screened out by the above methods to explore the driving mechanism of spatiotemporal variation of CAR in the study area, and in the selection of regression model, choose to use stepwise regression method, this model can solve the problem of population、 economics environment factors' multi-factor multicollinearity. The linear regression equation is shown in equation (4). The regression coefficients were calculated based on the data of the independent variables and the dependent variable by the least square method. The regression coefficients indicate the importance of each index. The regression equation reflects the relationship between the independent variables and the dependent variable.

$$y = a + b_1 x_1 + b_2 x_2 + \cdots + b_k x_k \tag{4}$$

where $a$ is a constant, $b$ is the regression coefficient, $x$ is the impact factor, and $y$ is the rate of cropland abandonment.

## 4 Results and discussion

### 4.1 Characteristics of spatial-temporal of abandoned cropland

Fig 2 shows the abandoned cropland area (ACA) and CAR during 2001–2020. From Fig 2B, it can be seen that there is a wave-like change in the abandoned cropland in the study area. In 2014, the largest ACA in the study area was abandoned, at 45.81 km², with a CAR of 14.15%; in 2001, the smallest ACA was 2,314.21 km², with a CAR of 7.68%. According to Fig 2A, there is a clear difference in ACA inhe three cities. Quijing city has the largest area of abandoned cropland each year, with the highest ACA in 2007, at 2,587.49 km², with a CAR of 62%.

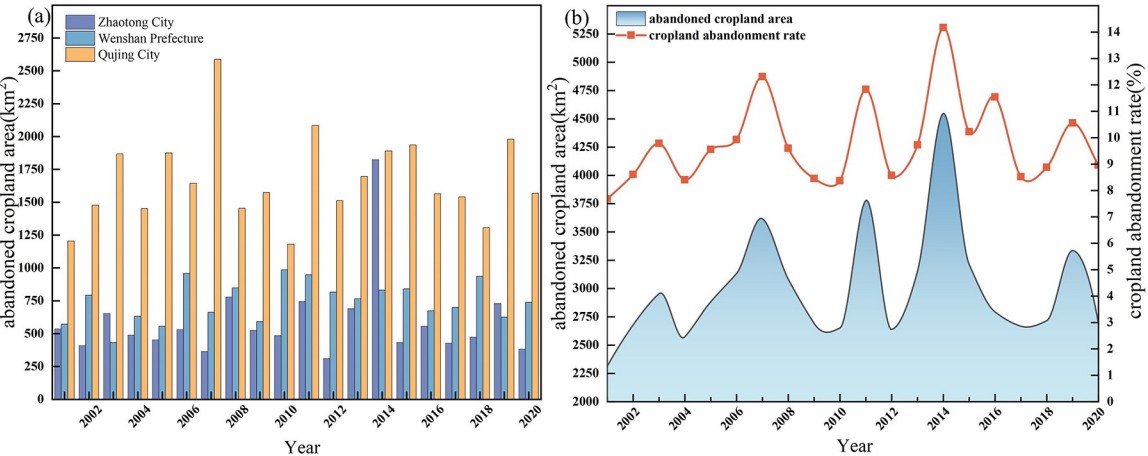

**Fig 2. The time variation of abandoned cropland area and cropland abandonment rate in KPEYC.** (a) The changes of the abandoned cropland area in each city. (b) The changes of the cropland abandoned area and the cropland abandonment rate in KPEYC in the past 20 years.

Wenshan prefecture has tended to stabilize its ACA over the past two decades, with an average of 747.32 km$^2$, with an average CAR of 11.44%. Zhaotong city had the largest ACA, with 1,825.07 km$^2$, with a CAR of 7.68%. Zhaotong city had the largest fluctuation in ACA, with the largest ACA in 2014, with 1,825.07 km$^2$ and CAR of 19.20%, and the smallest ACA in 2012, with 310.32 km$^2$ and CAR of 3.41%.

According to the quantity of the 20-year CAR of 28 counties in KPEYC, the abandonment degree of each county in each year is expressed by dividing it into 4 levels according to the equal interval method. The CAR were classified: class I (CAR ranging from 0% to 4.445%), class II (CAR ranging from 4.445% to 7.85%), class III (CAR ranging from 7.85% to 12%), and class IV (CAR ranging from 12% to 70%). As shown in Fig 3, Class I and II with an insignificant degree of cropland abandonment were mainly distributed in the northern and southern parts of the study area, while Class III and IV with serve cropland abandonment were distributed in the central and southern parts, showing a "high in the center and low in the north and south" distribution pattern of abandoned cropland. where rolling hills, river valleys, gullies, and ravines exist in the complex terrain, and rock desertification is severe. From 2001 to 2010 timeframe, the worst areas of cropland abandonment were spread over the whole of Qujing city and parts of Wenshan prefecture, with Zhaotong city having a lower level of cropland abandonment. From 2010 to 2015, the CAR in Wenshan prefecture continued to increase, the scale of the cropland abandonment further expanded, and after 2016, the cropland abandonment in the city decreased. Overall, it seems that in these 20 years, the most serious phenomenon of cropland abandonment was found in east-central Yunnan, with a large area and a deep degree of cropland abandonment, while the degree of cropland abandonment in the north was shallow, and the cropland abandonment situation in the south was extremely unstable, with large fluctuations in the degree of cropland abandonment in different counties. Most of the counties and districts with high CAR are located in areas with good economic conditions and easy transportation; low CAR is located in densely populated, economically backward areas with complex terrain and poor transportation.

## 4.2 Drivers of temporal changes of CAR

Due to the fact that certain attributes of driving factors in the natural environment remain constant over a long period, if correlation analysis is conducted, it cannot reflect their

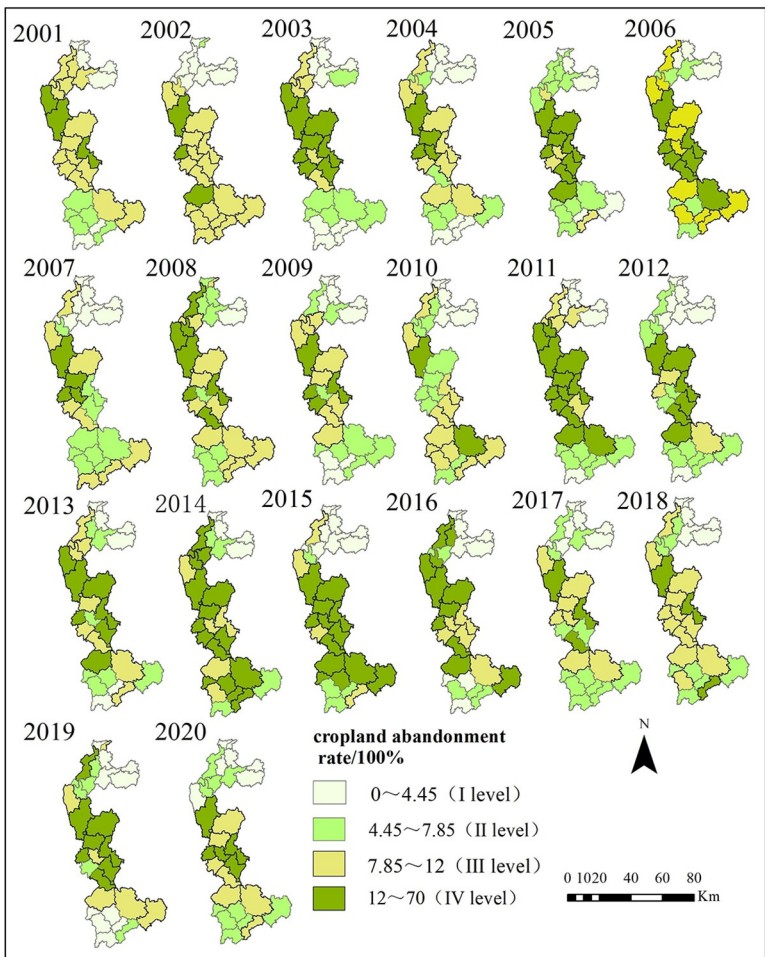

**Fig 3. Distribution of cropland abandonment rate at county scale in KPEYC during 2001–2020.**

correlation with the CAR. Therefore, the driving factors selected for Spearman correlation analysis are primarily those that vary over time. In this paper, we take the entire study area as the unit of calculation and use correlation analysis to explore the degree of correlation of population, economic environment, two farming conditions factors and changes in CAR. The results are shown in Fig 4. The variables examined are gross value of agricultural production (X5), gross value of industrial production (X6), gross tertiary industrial production (X7), and agriculture machinery power (X18). The correlation between economic factors and cropland abandonment is the largest, with $R_{X5}$ is 0.329, $R_{X6}$ is 0.302, and $R_{X7}$ is 0.274. Agriculture machinery power, among the farming conditions, has a correlation value of 0.289 with the change in the CAR. However, the $R_s$ between the population and the CAR are lower. This indicates that the direct role of people and cropland abandonment has no obvious relationship.

KPEYC's rugged and mountainous terrain, coupled with its poor soil properties, has resulted in extremely limited cropland and greatly restricted the scale of agricultural production. As a result, large-scale mechanized agricultural production is impossible, and the level of agricultural mechanization is among the lowest in the country. In response, local governments are vigorously developing the industrial and service sectors as a way to drive economic growth. However, as the cost of agricultural production continues to rise and relative efficiency

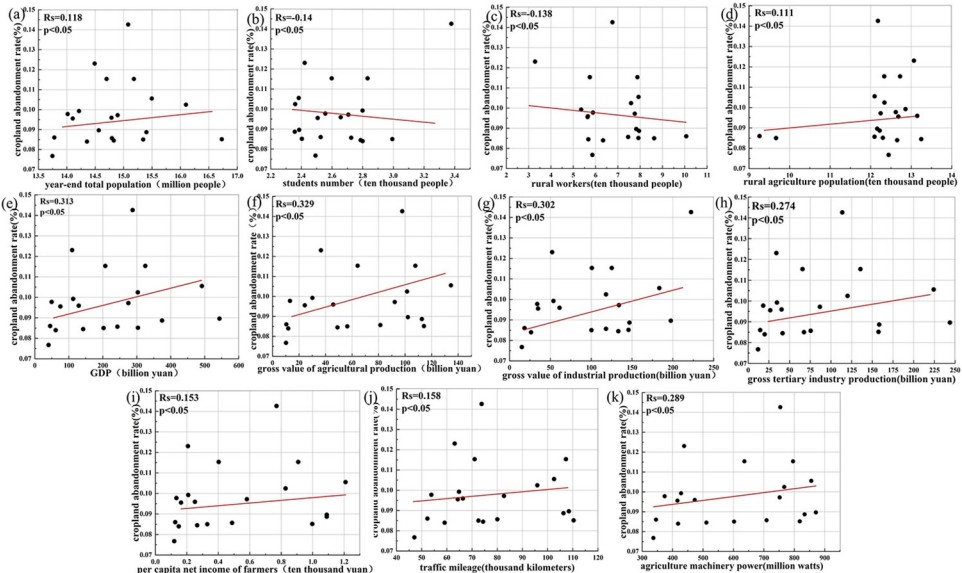

**Fig 4.** The scatter plot between the cropland abandonment rate and (a) year-end total population; (b) students number; (c) rural workers; (d) rural agriculture population; (e) GDP; (f) gross value of agricultural production; (g) gross value of industrial production; (h) gross tertiary industry production; (i) per capita net income of farmers; (j) traffic mileage; (k) agriculture machinery power.

remains low, the traditional subsistence economy of small farms is no longer able to meet the increasing demands of both production and daily life. As a result, farmers have shifted their focus to non-agricultural industries, leading to an increase in migrant labors and population out-migration. This has led to the cropland abandonment of large areas of cropland in the eastern region.

## 4.3 Drivers of spatial variation of CAR

Neglecting the time factors, all the factors and the CAR can be calculated as 20-year average data. To analyze the driving factors of the spatial distribution of abandoned cropland in the study area, geodetector model is used. When using geodetector methods, the numerical quantities of the factors must be transformed into categorical quantities. To ensure the reasonableness of the classification, through the review of relevant literature, the factors will be divided into 4–7 levels based on the size of the numerical value to be detected by reviewing the relevant literature.

The results of the analysis are shown in Table 4. As can be seen, the levels of the influencing factors are, in descending order, X7>X6>X9>X8>X14>X11>X18>X5>X17>X2>X12>X1>X3>X13>X10>X3>X16 with p-values of less than 0.05. All 18 selected drivers passed the significance test. In terms of the economic environment, gross tertiary industrial production (X7), gross value of industrial

**Table 4. Drivers detection results of the mean value in the past 20 years.**

| Factor | X1 | X2 | X3 | X4 | X5 | X6 | X7 | X8 | X9 |
|---|---|---|---|---|---|---|---|---|---|
| $q$ value | 0.189 | 0.216 | 0.171 | 0.114 | 0.264 | **0.430** | **0.445** | **0.334** | **0.360** |
| Factor | X10 | X11 | X12 | X13 | X14 | X15 | X16 | X17 | X18 |
| $q$ value | 0.136 | 0.284 | 0.199 | 0.151 | **0.317** | 0.240 | 0.086 | **0.235** | **0.269** |

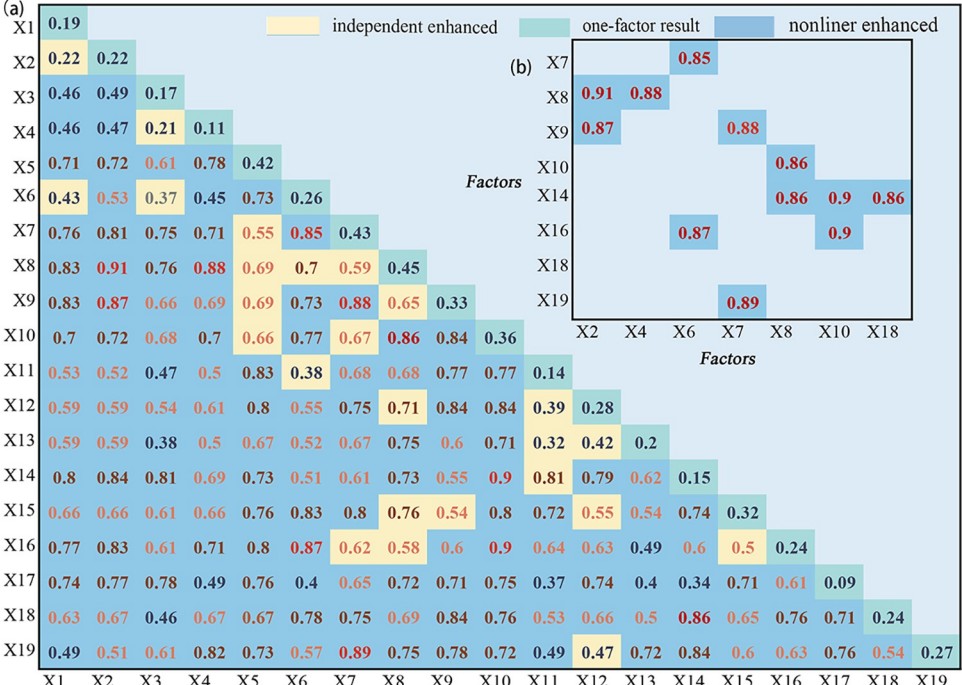

**Fig 5. The interactive detection results of the mean of the potential factors in the past 20 years.** (a) The interactive detection results of all potential factors. (b) High-value graph of interactive detection results.

production (X6) have the strongest explanatory power for the spatial distribution of cropland abandonment. These factors rank in the top two with q-values of 0.445, 0.430 respectively. In addition, the q-value of per capita net income of farmers (X8) is 0.334 and ranks third. The q-value of soil thickness (X9) among cropland attributes is 0.360, ranking fourth, while the q-value of elevation (X14) among farming conditions is 0.317, ranking fifth. As a result, the main driving factors of the spatial distribution of cropland abandonment are the gross tertiary industrial production, gross value of agricultural production, soil thickness, per capita net income of farmers, and elevation.

In terms of the economic environment, gross tertiary industrial production has the strongest explanatory power among the factors. This is due to the imbalance between urban and rural development caused by the rapid expansion of urbanization and industrialization, the inefficiency of food production, and the displacement of a large number of rural workers. In addition, cropland in KPEYC mainly consists of sloping land, making it difficult for the remaining rural worker groups to endure the long-distance, difficult, and high-intensity agricultural work. As a result, a significant amount of abandoned cropland. The explanatory power of soil thickness for cropland attributes ranks fourth. The thinner the soil thickness, the more likely it is to lead to soil erosion, a decrease in the efficiency of cultivation, and ultimately cropland abandonment by farmers who give up planting.

The results of the interaction detection are shown in Fig 5A. The results show that the influence of any two factors after their interaction is greater than the influence of these factors alone. Most of them show non-linear enhancement, while some of them show two-factor enhancement. From the Fig 5B, these three pairs with the strongest driving force after interaction are the number of students in primary (X2) and gross tertiary industrial production (X7) with a q-value of 0.91, the interaction of soil thickness (X9) and Average precipitation (X13) with a q-value of 0.90, and the soil thickness (X9) and slope (X15) with a q-value of 0.90.

Specifically, the spatial distribution of abandoned cropland in the whole region seems to be most influenced by the number of students in the family and the regional economy. In addition, poor cropland resources in the region and inconvenient farming conditions contribute to the development of cropland abandonment. The large number of students exacerbates the already heavy burden of the local economy on agricultural production, forcing farmers to seek non-agricultural work elsewhere to support their families [24]. Rapid economic development in some regions provides farmers with a large number of service-related jobs luring them to the city to work. Push and pull factors encourage the occurrence of cropland abandonment. Soil thickness decreases due to soil erosion, which is mainly caused by slope and precipitation. As a result, soil thickness shows nonlinear enhancement with slope and precipitation.

## 4.4 Drivers mechanism of spatial-temporal variation of cropland abandonment

After conducting correlation analysis and geodetector tests, we identified nine factors that have the strongest impact on cropland abandonment in KPEYC, including gross value of agricultural production (X5), gross value of industrial production (X6), gross tertiary industrial production (X7), per capita net income of farmers (X8), soil thickness (X9), elevation (X14), traffic mileage (X17), and agricultural machine power (X18). We conducted multiple linear stepwise regression analyses performed after normalization of factors and found that only six of these factorsX5, X7, X9, X14, and X17—had a significant impact on cropland abandonment. The CAR follows a normal distribution and the adjusted $R^2$ for our model is 0.532 with a standard deviation of 1.537, indicating a well-constructed model. Our regression estimation yields the following Eq (5):

$$y = 0.26 - 0.153X_5 + 0.148X_7 + 0.021X_9 - 0.133X_{14} - 0.031X_{17} \tag{5}$$

According to the regression results Table 5, all independent variables in the model passed the significance test at the 0.01 level, indicating their statistical significance. From the model, we can see that gross value of agricultural production, and gross tertiary industrial production among the economic environment factors have a significant effect on CAR. The results show that the higher the level of economic development of the whole region, the higher the CAR; while gross value of agricultural production is negatively correlated with the CAR. The coefficients of the remaining three variables, soil thickness, elevation, traffic mileage, are slightly lower at 0.021, -0.133, and 0.031, respectively. elevation and traffic mileage have a negative correlation with the CAR, and soil thickness has a positive correlation with the CAR. The cropland abandonment drive in East Yunnan has a strong complexity, i.e., cropland abandonment is likely to occur in areas with good economic development conditions, poor agricultural foundation, low level of tertiary industry development, relatively good soil conditions, low elevation, and inconvenient transportation conditions.

**Table 5. The results of multiple linear regression model.**

| Factor | ratio | standard error | T | significance | VIF |
|---|---|---|---|---|---|
| constant | 0.26 | 0.027 | 4.100 | 0.002 | |
| X14 | -0.133 | 0.013 | -9.039 | 0.000 | 1.015 |
| X5 | -0.153 | 0.034 | -7.045 | 0.000 | 1.026 |
| X9 | 0.021 | 0.008 | -3.022 | 0.000 | 1.027 |
| X17 | -0.031 | 0.013 | 2.284 | 0.004 | 1.015 |
| X7 | 0.148 | 0.34 | 4.323 | 0.000 | 1.246 |

The model and actual research indicate that the economic environment has the greatest impact on cropland abandonment, the rapid development of the region's economy will lead to a large number of workers leaving the land, which will lead to an increase in the CAR; In the selection of crops, due to the low efficiency of food production, KPEYC is the development of highland specialty crops, mainly planted in the baking tobacco, fruits, medicinal herbs, flowers, vegetables, and other cash crops, which are not suitable for cash crop growth of land farmers often choose to abandon. In the past decade, KPEYC has vigorously developed tourism to promote the development of the regional economy, and rural tourism is an important part of it. The government and tourism cooperatives have joined hands with individual farmers to actively create agricultural landscapes, such as rapeseed flower fields in Luoping county and pear and peach blossom festivals in Wenshan prefecture, which have increased farmers' enthusiasm for planting crops and reduced the CAR to some extent. The areas with relatively good soil conditions and gentle terrain in the study area are mainly located in the central region, which has a strong industrial base, a favorable economic environment, and more non-agricultural jobs, resulting in a higher abandonment rate. The CAR is closely related to transportation conditions; the worse the transportation conditions are, the more inconvenient it is to cultivate, and the higher the CAR will be.

## 5 Discussion

### 5.1 Implications and generalizability of the results

The research conducted on the issue of CAR in KPEYC reveals that economic factors play a pivotal role in influencing this phenomenon. These economic factors directly impact the reduction in agricultural income and the increase in farmers' consumption levels, thereby affecting their decision to abandon their lands. This finding resonates with the principles outlined in von Thünen's rent theory [25, 26], which posits that the marginalization of land due to its poor suitability and diminishing economic returns underlie the decision to abandon land. However, it diverges from the findings of Han Ze [27], who highlighted that cropland abandonment in the karst mountains of Guizhou-Guangxi, China, is predominantly driven by agricultural practices and soil conditions, with soil erosion being a critical determinant of its distribution. The discrepancy could be attributed to the regional variations in the spatial distribution of cropland abandonment or possibly influenced by the different stages of regional economic development; for instance, Yunnan Province has exhibited higher GDP outputs compared to Guizhou Province over the past years.

The methodology employed in this research can indeed be applied across diverse regions globally; however, it is essential to recognize that varying data sets or distinct political cultures may yield different outcomes. For instance, when applied to countries outside of China, discrepancies in land management systems could potentially result in varying primary factors identified through the analysis.

### 5.2 Policy recommendations for preventing and controlling cropland abandonment

Based on the findings of this study and in alignment with the management policies of the Yunnan Province area in China, the following recommendations are proposed to effectively tackle the issue of cropland abandonment and ensure food security in the region. To address the labor outflow issue induced by economic development, the following measures are suggested that: (i) Rural agricultural subsidies should be enhanced: This would provide much-needed financial support to farmers, especially those cultivating land in mountainous regions. (ii)

Preferential treatments should be extended to farmers in marginal lands: For instance, by offering educational and medical benefits to the children of these farmers, thus alleviating the economic burden on their families. (iii) Financial risk mitigation for farmers: By providing low-interest loans and agricultural insurance, farmers in marginal mountainous areas can better withstand financial risks associated with agricultural production. (iv) Expanding sales channels through internet platforms: Utilizing online platforms to open up new sales channels and expand markets for agricultural produce, especially those that are less amenable to long-distance transportation, thereby reducing the sales risk for farmers. (v) Enhanced agricultural technology support: By improving seeds, providing advanced agricultural cultivation technologies, and boosting crop productivity.

For the severely rocky desertified areas in KPEYC, it is imperative to thoroughly implement the concept of an ecological civilization. This includes executing afforestation projects on mountaintops, developing specialized economic industries on the slopes, and undertaking the "slope-to-staircase" project to fully utilize arable land and enhance land productivity. Such measures not only conserve water resources but also reduce soil erosion and tackle pest issues.

While governments at all levels have made preliminary progress in addressing the issue of abandoned cropland, the protection of cropland resources remains a long-term process aimed at promoting the rational use of these resources. To continuously stimulate farmers' enthusiasm for farming, it is necessary to establish a long-term early warning and remediation mechanism for abandoned cropland to prevent the recurrence of this phenomenon.

## 5.3 Study limitations

This research analyzes the influencing factors and possible mechanisms of cropland abandonment using correlation coefficients and geodetector, and achieves certain results. However, there are some limitations that may introduce uncertainties into the results. First of all, the data used in this study were obtained from statistical yearbooks and remote sensing products. These data may have some errors, which could lead to inaccuracies in the results. When extracting abandoned cropland, there may be a mixed pixel that is still recognized as abandoned plots, resulting in an overestimation of the actual area of abandoned cropland. This could have a certain impact on the results of the study. Second, the analysis is limited to the county level, and all indicators are the result of county-wide statistics, which could ignore internal differences. In the future, it would be interesting to explore the impact factors of cropland abandonment at smaller spatial scales. Finally, the results of the geodetector method are influenced by the way spatial partitioning is done, but there is currently no widely recognized and reliable method for spatial partitioning. It would be necessary to conduct quantitative research on the effects of spatial partitioning on geodetector in the future.

## 6 Conclusion

To analyze the spatial-temporal patterns in KPEYC, this study used correlation analysis and geodetector to identify the main factors that have driven the cropland abandonment in this region over time and space. By applying linear regression analysis to explore the driving mechanisms behind the spatial-temporal changes in cropland abandonment. The following conclusions were reached:

1. The CAR in KPEYC shows a fluctuating trend, with an average value of 9.78%, and a spatial distribution pattern of "high in the center and low in the north and south". From 2010 to 2015, the most prominent phenomenon of cropland abandonment occurred, and the

situation improved in Wenshan Prefecture after 2015. Compared to the other two areas, Zhaotong City has a lower CAR.

2. In terms of temporal changes, the economic environment and farming conditions have obvious correlations with cropland abandonment gross value of agricultural production, and gross value of industrial production in the economic environment have the largest correlation coefficients with CAR, and mechanical power has the highest correlation with changes in abandonment rates in farming conditions.

3. Gross tertiary industrial production, gross value of industrial production has the strongest explanatory power for the spatial distribution of cropland abandonment, followed by soil thickness.

4. Gross value of agricultural production, and gross tertiary industrial production are the main driving factors of spatial and temporal changes of CAR; the higher the gross tertiary industrial production, the higher the CAR. Elevation, soil thickness, and traffic mileage are the main driving factors of the spatial and temporal changes of the CAR.

## Acknowledgments

Authors would like to thank all colleagues and teachers in the laboratory for their generous help in our experiments. The authors would also like to thank the anonymous reviewers for their very competent comments and helpful suggestions.

## Author Contributions

**Formal analysis:** Jianhong Xiong.

**Investigation:** Mengzhu Sun.

**Methodology:** Jiasheng Wang.

**Software:** Yongchao Ma.

**Writing – original draft:** Jingyi Wang.

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
