## [Decision Letter · Decision Letter 0]

14 Feb 2024

PONE-D-23-43715Spatial-temporal characterization of cropland abandonment and its driving mechanisms in the Karst Plateau in Eastern Yunnan, China, 2001-2020PLOS ONE

Dear Dr. Wang

Thank you for submitting your manuscript to PLOS ONE. After careful consideration, we feel that it has merit but does not fully meet PLOS ONE’s publication criteria as it currently stands. Therefore, we invite you to submit a revised version of the manuscript that addresses the points raised during the review process.

We look forward to receiving your revised manuscript.

Kind regards,

Eda Ustaoglu, PhD

Academic Editor

PLOS ONE

Journal Requirements:

"This work was supported by the National Natural Science Foundation of China (Grants 41961056); Yunnan Province high-level Talent Training Support Program (Grant No. YNWR-QNBJ-2020-053); Yunnan Normal University 2023 Graduate Student Research and Innovation Fund (Grant No. YJSJJ23-B149)."

Please state what role the funders took in the study.  If the funders had no role, please state: ""The funders had no role in study design, data collection and analysis, decision to publish, or preparation of the manuscript."" If this statement is not correct you must amend it as needed. 

3. Please be informed that funding information should not appear in the Acknowledgments section or other areas of your manuscript. We will only publish funding information present in the Funding Statement section of the online submission form. Please remove any funding-related text from the manuscript. 

5. We note that Figure 1 and 3 in your submission contain map/satellite images which may be copyrighted. All PLOS content is published under the Creative Commons Attribution License (CC BY 4.0), which means that the manuscript, images, and Supporting Information files will be freely available online, and any third party is permitted to access, download, copy, distribute, and use these materials in any way, even commercially, with proper attribution. For these reasons, we cannot publish previously copyrighted maps or satellite images created using proprietary data, such as Google software (Google Maps, Street View, and Earth). For more information, see our copyright guidelines: http://journals.plos.org/plosone/s/licenses-and-copyright.

(1) You may seek permission from the original copyright holder of Figure 1 and 3 to publish the content specifically under the CC BY 4.0 license.  

**Additional Editor Comments:**

The three reviewers advise for a major revision while the other one suggests a minor revision. You can find the details of advised revisions in the report of the reviewers that are attached to this e-mail. We expect you to provide an answer to these comments and address the points raised by the reviewers.

Reviewers' comments:

Reviewer's Responses to Questions

**Comments to the Author**

1. Is the manuscript technically sound, and do the data support the conclusions?

Reviewer #1: Yes

Reviewer #2: Partly

Reviewer #3: Yes

Reviewer #4: Partly

2. Has the statistical analysis been performed appropriately and rigorously? 

Reviewer #1: Yes

Reviewer #2: No

Reviewer #3: Yes

Reviewer #4: Yes

3. Have the authors made all data underlying the findings in their manuscript fully available?

Reviewer #1: Yes

Reviewer #2: Yes

Reviewer #3: Yes

Reviewer #4: Yes

4. Is the manuscript presented in an intelligible fashion and written in standard English?

Reviewer #1: Yes

Reviewer #2: No

Reviewer #3: Yes

Reviewer #4: No

5. Review Comments to the Author

Reviewer #1: General comments:

The research has complete comprehensiveness about the spatial and temporal analysis and its mechanisms of abandoned agricultural lands in the Yunnan Plateau in China for the period 2001-2020, which indicates that a great effort has been made in completing the research, analyzing its variables, and producing good results about the phenomenon of abandonment of agricultural lands in that region, and we point out some of the following observations:

- Identifying agricultural lands in the Yunnan Plateau.

- Adding recent scientific papers on the abandonment of agricultural lands in China and in that region by combining temporal and spatial studies on the abandonment of agricultural lands.

Line 46: The topic is vital and a recommendation must be added to the research based on the results obtained

Line 66: reference #9 missed

Line 147: add i at the end “the year before the year i".

Line 176: in the methods, the author did not explain the test used in testing the time series

Line 188: in the Economic Environment variables, the problem of linear correlation usually appears in such variables. The author did not indicate here the reason for adding these variables, knowing the existence of this problem.

Line 188: There are about 18 previous studies that were mentioned on pages 3, 4, and 5, and the researcher must link them to the description.

Line 249: What does “ACR” mean?

Line 265: How were these proportions divided? Or what method is used in division?

Line 287: why the # of drivers decrease from 19 to 11 (author didn’t mention the rezone).

Line 290-292: Do not forget correlation sign…

Line 337: the result of X4 and X19 not equal to the numbers in figure 5.

Line 355: there were some variables had high-value interaction detection not mention, such as X4 with X8 and X14 with X18.

Line 375: what does “o” mean?

Line 400: change the word “thesis” to “study”.

Line 410: 9.78%

Line 438: what does “KPTYC” mean?

Line 461: Cereal were not discussed in the research, so why did the result appear here? Note that the Cereal area is large.

Line 513: reference #9 not mention in the study.

Reviewer #2: Lack of clarity in research objectives: The article does not clearly state the specific research objectives or hypotheses that the study aims to address.

Insufficient methodological description: The article briefly mentions the use of correlation analysis, geodetector, and regression analysis methods, but it lacks a detailed description of how these methods were applied.

Inadequate data presentation: The article mentions the selection of 19 key factors from population, economic environment, cropland attributes, and farming conditions, but it does not provide a clear explanation of the rationale behind the selection of these factors. Additionally, the article lacks specific data on the values and trends of the selected factors, making it difficult to assess their impact on cropland abandonment.

Lack of discussion on limitations: The article does not acknowledge or discuss the limitations of the study. Every research project has inherent limitations, and it is important to address them to provide a balanced perspective. This omission reduces the credibility and reliability of the research findings.

Limited generalizability: The study focuses specifically on the Karst Plateau in Eastern Yunnan, China, and the findings may not be applicable to other regions or contexts. The article does not discuss the potential implications or generalizability of the findings, limiting the broader relevance of the research.

Absence of policy recommendations: Despite mentioning that the study can provide decision support for preventing and controlling cropland abandonment, the article does not provide any specific policy recommendations or practical implications based on the research findings. This omission reduces the practical applicability and impact of the study.

Reviewer #3: The article titled "Spatial-temporal characterization of cropland abandonment and its driving mechanisms in the Karst Plateau in Eastern Yunnan, China, 2001-2020" presents a comprehensive analysis of the phenomena of cropland abandonment in the Karst Plateau in Eastern Yunnan, China (KPEYC) over two decades. The outlines the study's objectives, methodologies, and findings. Here are some comments on the article:

Scope and Relevance: The study addresses a significant and pressing issue of cropland abandonment, which has implications for regional food security in a challenging geographical and economic context. The focus on the KPEYC, characterized by its mountainous terrain and low mechanization, highlights the unique challenges faced in such environments. The temporal scope of 2001-2020 allows for an analysis of trends over a substantial period, enhancing the study's relevance.

Justification of the statement: Page 4, paragraph 2. The given statement must have to justify with previous studies [1-3] as “The survey method is suitable for discovering the driving factors of geographical phenomena [1,2]”

[1] https://doi.org/10.1016/j.jag.2022.102936

[2] https://doi.org/10.1016/j.jag.2022.102936

[3] https://doi.org/10.1016/j.jag.2022.102936

Methodological Robustness: The application of correlation analysis, geodetector, and regression analysis methods from three perspectives (temporal change, spatial distribution, and spatial-temporal change) indicates a robust methodological approach. Selecting 19 key factors across diverse domains (population, economic environment, cropland attributes, and farming conditions) suggests a comprehensive attempt to understand the multifaceted nature of cropland abandonment. However, the artilce does not detail the criteria for selecting these factors or their individual contributions, which would be crucial for assessing the study's analytical depth.

Findings and Interpretations: The identification of fluctuating trends in the cropland abandonment rate (CAR) and the spatial distribution pattern provides valuable insights into the issue's complexity. The linkage of economic factors (GDP, gross value of agricultural and industrial production) with CAR is particularly noteworthy, indicating the significant impact of economic transitions on agricultural practices. While the study highlights the core driving forces and contributing factors to CAR, it could benefit from a more nuanced discussion on how these factors interact and the potential for mitigating negative impacts through policy interventions.

Contributions and Implications: The study's findings contribute to the understanding of land use changes in karst regions and the critical role of economic factors in driving these changes. By pinpointing specific economic indicators as decisive factors, the research offers a foundation for targeted policy measures aimed at balancing economic development with sustainable land use practices. However, the article does not explicitly discuss the implications for local communities, food security, and ecological sustainability, which are critical for translating research findings into actionable strategies.

Limitations and Future Research: While the article indicates a comprehensive analysis, it does not address the limitations of the study, such as potential biases in data selection, the resolution of spatial data, or the generalizability of the findings to other karst regions. Future research directions could include exploring the socio-economic impacts of cropland abandonment on local communities, integrating ecological considerations into economic analyses, and developing adaptive management strategies that address both economic and environmental sustainability.

Reviewer #4: The manuscript is well-written, however, the discussion part should be polished by comparing the findings with the existing studies. The conclusion should be shorten and be very specific. The figures came with low resolution and not understandable. Please upload high-resolution figures with the revised version of the manuscript.

6. PLOS authors have the option to publish the peer review history of their article (what does this mean?). If published, this will include your full peer review and any attached files.

Reviewer #1: No

Reviewer #2: **Yes: **Tsegamariam Dula Sherka

Reviewer #3: No

Reviewer #4: No

---

## [Author Response · Author response to Decision Letter 0]

9 Apr 2024

Dear Editor:

We really appreciate all of the insightful and constructive comments from the reviewers. We have thoroughly revised the manuscript (PONE-D-23-43715) according to all comments received. In our point-by-point response attached below, reviewers’ comments is in larger italic fonts (12 pt.) and our response is in smaller fonts (10 pt.). We hope that the revised version would be closer for publication in your journal. Look forward to hearing from you. Thank you.

With best wishes,

Yours sincerely,

Jiasheng Wang

 

Response to reviewers’ comments on our manuscript

PONE-D-23-43715

Reviewer #1

1. Identifying agricultural lands in the Yunnan Plateau.

Reply:

We've added information about agricultural lands in the Yunnan Plateau in Line 134 of the modified manuscript. The added content is as follows.

Added content:

The cropland area in KPEYC accounts for 35% of the province's cropland.

2. Adding recent scientific papers on the abandonment of agricultural lands in China and in that region by combining temporal and spatial studies on the abandonment of agricultural lands.

Reply:

Thanks for your suggestion. We added references in line 102. The supplementary references are as follows. 

Added content:

Line 102,

From the perspective of research methods, the existing research on cropland abandonment lacks an analysis of the driving mechanism of spatial and temporal changes. From the perspective of driving factors, existing studies mainly analyze the driving mechanisms of cropland abandonment in terms of parcels or households[19,20,21].

[19] HAN Dejun., YAN Zhao., SONG Yang. Study on the two-way effect of "alleviating abandonment" and "intensifying non-food" of agricultural land system innovation--The example of "three changes" reform in southwest mountainous area[J]. China Land Science,2023,37(09):73-84.

[20] QUAN Zhijun, WEN Jun, LI Pengju. Research on the status quo and countermeasures of arable land protection in the context of food security--Taking Yibin City as an example[J]. Southern Agricultural Machinery,2023,54(24):175-177.

[21] YANG Ruixue,PENG Haiying. A review of domestic and international research on cropland abandonment[J]. Journal of Management Cadre College of Ministry of Agriculture and Rural Affairs,2023,14(03):59-65. 

3. Line 46: The topic is vital and a recommendation must be added to the research based on the results obtained

Reply:

Thank you for your review and comments. In this paper, we have changed it according to your suggestion in line 45-51.

modified content:

Based on the results, this study can provide decision-making support for local prevention and control of cropland abandonment, and the local community needs to promote land transfer and concentration and local urbanization according to local conditions, improve agricultural policies, improve farming conditions, etc. in order to increase farmers' enthusiasm for production, promote the rational use of cropland, and solidly push forward ecological restoration and management, optimize ecological spatial patterns, manage serious areas of rocky desertification, and appropriately alleviate the contradiction between people and land.

4. Line 66: reference #9 missed.

Reply:

Thank you for your review and comments. References #9 have been added in line 71 of the modified manuscript.

modified content: 

Scholars at home and abroad have conducted extensive research on the driving mechanisms and influencing factors, mainly analyzing the factors of abandonment from the perspective of the social system and land use change[8,9,10,11].

5. Line 147: add i at the end “the year before the year i”;

Reply:

Thanks for your valuable counsel. We have added the add i at the end. The modified content is as follows:

modified content: 

where is the CAR in the year , is the area of cropland abandonment in the year , and is the area of cropland in the year before the year .

6. Line 176: in the methods, the author did not explain the test used in testing the time series. 

Reply:

Thank you again for reviewing the manuscript so carefully. In response to the reviewer's questions regarding the time series testing method, we have now detailed the ADF test used in the methods section, including the purpose of the test, steps, and parameter settings. We believe these supplements will help readers better understand our research methods and results.

added content:

3.2 Time series testing

To test the stationarity of the abandoned cultivated land over time series, we employ the ADF (Augmented Dickey-Fuller test), a rigorous statistical testing method. It determines whether a time series is stationary by comparing the test statistic with its corresponding critical values. If the test statistic is less than the critical value and the p-value is less than 0.05, we can reject the null hypothesis and conclude that the time series is stationary. Results are shown in Table 2. As can be seen from Table 2, when the difference is at order 0, the significance P-value is 0.001, which indicates significance at the given level. Thus, the null hypothesis is rejected, and the series is considered to be a stationary time series.

Table 2. ADF Time Series Test Results Table

variable Difference Order t P AIC 1% 5% 10%

CAR 0 -4.132 0.001 -56.957 -3.833 -3.031 -2.656

 1 -5.414 0.000 -51.999 -3.889 -3.054 -2.667

 2 -0.904 0.787 -50.326 -4.332 -3.233 -2.749

7. Line 188: in the Economic Environment variables, the problem of linear correlation usually appears in such variables. The author did not indicate here the reason for adding these variables, knowing the existence of this problem.

Reply:

Thank you for your comments the manuscript so carefully. The reason for selecting these variables is that these variables are directly related to cropland abandonment, and their relationship with cropland abandonment has been clearly explained in the Description column of Table 1. In the revised version, a table description has been added at the end of the last first paragraph of section 3.1. There is indeed a problem of linear correlation between economic environment factors. GDP is the sum of output value of the three industries, so the GDP factor has been removed from the revised draft, and the result has also been modified accordingly. There was no obvious linear correlation between the other variables.

8. Line 188: There are about 18 previous studies that were mentioned on pages 3, 4, and 5, and the researcher must link them to the description.

Reply:

Thank you for your comments the manuscript so carefully. In this paper, we mentioned here is that have referred to many references earlier, so in the Description section of Table 1, We linked to these references.

modified content:

Level

Indicators Level 2

 Indicators Unit Description

Population Year-end total population X1

 Million people

Million people

Million people

Million people The larger the population, the less likely it is that cropland will be left fallow. [1]

 Number of students in primary and secondary schools X2 It can be used as supplementary labor for farming. [14]

 Rural workers X3

 The lower the amount of labor, the higher the likelihood of abandonment. [5]

 Agricultural population X4

 A decrease in the rural agricultural population makes abandonment more likely. [1]

Economics

Environment Gross value of agricultural production X5 Billion yuan

Billion yuan 

Billion yuan

Ten thousand yuan The greater the value of agricultural production, the more likely it is to be abandoned. [4]

 Gross value of industrial production X6 The more developed the industry is, the more likely it is to be abandoned. [7]

 Gross tertiary industrial production X7 The larger the output value of tertiary industry is, the more likely it is to be abandoned. [13].

 Per capita net income of farmers X8 The higher the income of farmers, the higher the likelihood of abandonment. [5]

 Cropland

Attributes Soil thickness X9 Meter The thicker the soil layer, the less likely it is to be abandoned

 Soil organic matter content X10 Grams per kilogram The higher the quality of the soil, the less likely it is to be abandoned. [5]

 Average sunshine duration X11 Hour The better the light, the less likely it is to be abandoned.

 Average air temperature X12 Celsius The better the temperature, the less likely to be abandoned.

 Average precipitation X13 Millimeter The richer the water resources, the less likely it is to be abandoned.

Farming

Conditions Elevation X14 Kilometer The higher the altitude, the more prone to abandonment. [14]

 Slope X15 Degree The steeper the slope, the more likely it is to be abandoned. [11]

 Slope direction X16 Degrees The sunnier the slope, the better the light, the less likely it is to be abandoned. [14]

 Traffic mileage X17 Thousand Kilometers Poor access to cropland can lead to abandonment. [11]

 Agriculture machinery powerX18 Million Watts The lower the degree of mechanization, the higher the likelihood of abandonment. [5]

9. Line 249: What does “ACR” mean?

Reply:

Thank you for your detailed comments. I am sorry, this word is my spelling error, the correct word is CAR.

10. Line 265: How were these proportions divided? Or what method is used in division?

Reply:

Thank you so much for your careful reading. The CAR were divided into four classified: class I (CAR ranging from 0% to 4.445%), class II (CAR ranging from 4.445% to 7.85%), class III (CAR ranging from 7.85% to 12%), and class IV (CAR ranging from 12% to 70%). These proportions are divided according to the following content.

added content:

According to the quantity of the 20-year CAR of 28 counties in KPEYC, the abandonment degree of each county in each year is expressed by dividing it into 4 levels according to the equal interval method.

11. Line 287: why the # of drivers decrease from 19 to 11 (author didn’t mention the reason).

Reply：

Thank you so much for your careful reading. The reason for the reduction in the number of driving factors from 19 to 11 is mainly due to the fact that some attributes of the driving factors in the natural environment remain constant over a long period. If correlation analysis is performed, it cannot reflect their correlation with the rate of land abandonment. Therefore, the driving factors selected for Spearman correlation analysis are primarily those that vary over time.

modified content:

Due to the fact that certain attributes of driving factors in the natural environment remain constant over a long period, if correlation analysis is conducted, it cannot reflect their correlation with the CAR. Therefore, the driving factors selected for Spearman correlation analysis are primarily those that vary over time. In this paper, we take the entire study area as the unit of calculation and use correlation analysis to explore the degree of correlation of population, economic environment, two farming conditions factors and changes in CAR.

12. Line 290-292: Do not forget correlation sign..

Reply:

Thank you so much for your careful reading. We'll modify it with the suggestions you gave us.

modified content:

The results are shown in Figure 4. The variables examined are gross value of agricultural production (X5), gross value of industrial production (X6), gross tertiary industrial production (X7), and agriculture machinery power (X18). The correlation between economic factors and cropland abandonment is the largest, with is 0.329, is 0.302, and is 0.274. Agriculture machinery power, among the farming conditions, has a correlation value of 0.289 with the change in the CAR.

13. Line 337: the result of X4 and X19 not equal to the numbers in figure 5.

Reply:

Thanks for your suggestion. Due to my carelessness, there are errors in the data in Figure 5. The figure is modified as follows. 

modified content:

14. Line 355: there were some variables had high-value interaction detection not mention, such as X4 with X8 and X14 with X18.

Reply:

Thanks for your suggestion. Due to my carelessness that I did not analyze these two pairs of interaction detection results. The relevant modifications have been highlighted in Figure 5, and since there are many combinators with large values of interaction detection q, the analysis in the text focuses on combinators with interaction factor values greater than 0.9. 

15. what does“o”; mean?

Reply:

Thanks for your suggestion. I typed an extra "o" because of my carelessness. 

modified content:

From the model, we can see that GDP, gross value of agricultural production, and gross tertiary industrial production among the economic environment factors have a significant effect on CAR.

16. change the word “thesis”; to “study”. 

Reply:

Thanks for your suggestion. I change the word thesis to study. 

modified content:

To analyze the spatial-temporal patterns in KPEYC, this study used correlation analysis and geodetector to identify the main factors that have driven the cropland abandonment in this region over time and space. By applying linear regression analysis to explore the driving mechanisms behind the spatial-temporal changes in cropland abandonment. The following conclusions were reached:

17. Line 410: 9.78%. 

Reply:

Thanks for your suggestion. Taking into account your comments and those of the other reviewers, we have revised this section，and deleted 9.78%.

18. Line 438: what does “KPTYC” mean? 

Reply:

Thanks for your suggestion. I change the KPTYC to KPEYC. 

modified content:

To analyze the spatial-temporal patterns in KPEYC, this study used correlation analysis and geodetector to identify the main factors that have driven the cropland abandonment in this region over time and space. By applying linear regression analysis to explore the driving mechanisms behind the spatial-temporal changes in cropland abandonment. The following conclusions were reached:

19. Cereal were not discussed in the research, so why did the result appear here? Note that the Cereal area is large. 

Reply:

Thanks for your question. Cereal is grain production in the research, and the last section is the policy recommendations made, fully incorporating your suggestions, and we have placed the policy recommendations section in the discussion section.

modified content:

According to the results of this study, it can still help the relevant departments to understand the status quo of regional cropland abandonment and provide corresponding data support for future cropland protection policies. It is necessary to create a different kind of countryside for “agriculture+ tourism”, promote land transfer and concentration according to local conditions, local urbanization, improve agricultural policies, and improve farming conditions in order to increase farmers' enthusiasm for production and promote the rational use of cropland; it is necessary to solidly push forward ecological restoration and governance, optimize the ecological spatial pattern, manage the serious areas of rocky desertification, and appropriately alleviate the contradiction between people and land. To establish a long-term mechanism to rectify the abandonment of cropland; to orderly promote the transfer of land in accordance with the law, to guide farmers who are not in a position to carry out normal farming to transfer land to the main body of the operation, and to encourage conditional places to carry out the market operation of land transfer, to revitalize the cropland resources, and to avoid the waste of land resources; at present, the governments of various levels of the abandonment of cropland have had a preliminary understanding of the rectification of cropland abandonment has had initial results, but the protection of cropland resources is a long-term process, so as to promote the rational use of cropland. The protection of cropland resources is a long-term process, to fully mobilize the enthusiasm of farmers to grow food, it is necessary to establish a long-term early warning of cropland abandonment, remediation mechanism, to put an end to the phenomenon of cropland abandonment of the resurgence of the phenomenon.

20. Line 513: reference #9 not mention in the study.

Reply:

Thank you for your review and comments. References #9 have been added in line 71 o

---

## [Decision Letter · Decision Letter 1]

16 May 2024

PONE-D-23-43715R1Spatial-temporal characterization of cropland abandonment and its driving mechanisms in the Karst Plateau in Eastern Yunnan, China, 2001-2020PLOS ONE

Dear Dr. Wang,

Thank you for submitting your manuscript to PLOS ONE. After careful consideration, we feel that it has merit but does not fully meet PLOS ONE’s publication criteria as it currently stands. Therefore, we invite you to submit a revised version of the manuscript that addresses the points raised during the review process.

There are minor issues need to be revised as suggested by the reviewers. You can address the reviewer's comments and re-submit a revised manuscript to be re-considered.

We look forward to receiving your revised manuscript.

Kind regards,

Eda Ustaoglu, PhD

Academic Editor

PLOS ONE

Journal Requirements:

Reviewers' comments:

Reviewer's Responses to Questions

**Comments to the Author**

1. If the authors have adequately addressed your comments raised in a previous round of review and you feel that this manuscript is now acceptable for publication, you may indicate that here to bypass the “Comments to the Author” section, enter your conflict of interest statement in the “Confidential to Editor” section, and submit your "Accept" recommendation.

Reviewer #1: All comments have been addressed

Reviewer #4: All comments have been addressed

2. Is the manuscript technically sound, and do the data support the conclusions?

Reviewer #1: Yes

Reviewer #4: Yes

3. Has the statistical analysis been performed appropriately and rigorously? 

Reviewer #1: Yes

Reviewer #4: Yes

4. Have the authors made all data underlying the findings in their manuscript fully available?

Reviewer #1: (No Response)

Reviewer #4: Yes

5. Is the manuscript presented in an intelligible fashion and written in standard English?

Reviewer #1: Yes

Reviewer #4: Yes

6. Review Comments to the Author

Reviewer #1: (No Response)

Reviewer #4: Dear Authors,

Thank you for the revisions made to the manuscript. The additions of new data, detailed methodological explanations, and comprehensive corrections show significant improvement and responsiveness to previous comments. However, I recommend a few further revisions to enhance the manuscript’s clarity and academic rigor:

1. Variable Selection Rationale: Please elaborate on how the selected variables directly contribute to addressing the study’s objectives and hypotheses. This would strengthen the logical flow and foundational basis of your analysis.

2. Limitations Discussion: Expand the discussion of the study's limitations, addressing potential biases and data resolution issues explicitly. This will add to the credibility and scholarly integrity of your findings.

3. Data Presentation: Ensure all figures are of high resolution and data presentations are accessible, aiding in the clear communication of your results and analyses.

4. Implications and Generalizability: Discuss the broader implications of your findings, particularly how they might apply to other regions or contexts, to enhance the manuscript's relevance and impact.

5. Policy Recommendations: Provide detailed policy recommendations based on your findings to demonstrate the practical applications of your research in addressing cropland abandonment.

7. PLOS authors have the option to publish the peer review history of their article (what does this mean?). If published, this will include your full peer review and any attached files.

Reviewer #1: No

Reviewer #4: No

---

## [Author Response · Author response to Decision Letter 1]

25 Jun 2024

1. Variable Selection Rationale: Please elaborate on how the selected variables directly contribute to addressing the study’s objectives and hypotheses. This would strengthen the logical flow and foundational basis of your analysis.

Reply:

We've added information about the variable selection rationale in line 186. The content outlines the process of selecting impact factors for the study of cropland abandonment.

Added content:

Cropland abandonment results from human utilization of cropland, which in turn is influenced by human decision-making. Making informed decisions requires a comprehensive consideration of both human and land factors. Human factors such as labor force and economic income play a crucial role in determining whether or not to abandon cropland, while land factors such as the attributes of farmland and farming conditions influence the location of abandonment. Given that this paper's research unit is the county, we have selected population, economic environment, cropland attributes, and farming conditions as the key influencing factors to be analyzed. 

2. Limitations Discussion: Expand the discussion of the study's limitations, addressing potential biases and data resolution issues explicitly. This will add to the credibility and scholarly integrity of your findings. 

Reply:

Thanks for your suggestion. We added a limitations discussion in line 484. This section discusses the study's methodology, data, and research scale limitations. The content is as follows. 

Added content:

This research analyzes the influencing factors and possible mechanisms of cropland abandonment using correlation coefficients and geodetector, and achieves certain results. However, there are some limitations that may introduce uncertainties into the results. First of all, the data used in this study were obtained from statistical yearbooks and remote sensing products. These data may have some errors, which could lead to inaccuracies in the results. When extracting abandoned cropland, there may be a mixed pixel that is still recognized as abandoned plots, resulting in an overestimation of the actual area of abandoned cropland. This could have a certain impact on the results of the study. Second, the analysis is limited to the county level, and all indicators are the result of county-wide statistics, which could ignore internal differences. In the future, it would be interesting to explore the impact factors of cropland abandonment at smaller spatial scales. Finally, the results of the geodetector method are influenced by the way spatial partitioning is done, but there is currently no widely recognized and reliable method for spatial partitioning. It would be necessary to conduct quantitative research on the effects of spatial partitioning on geodetector in the future.

3. Data Presentation: Ensure all figures are of high resolution and data presentations are accessible, aiding in the clear communication of your results and analyses.

Reply:

Thank you for your review and comments. We have uploaded high-resolution figures with the revised version of the manuscript. The resolution of the modified is 600 dpi.

modified content:

Fig. 1. The location of the study area. (a) The topographic map. (b) The KPEYC in China. (c) The 20-year change of total grain output and GDP.

Fig. 2. Distribution of abandoned cropland area and cropland abandonment rate. (a) The changes of the abandoned cropland area in each city. (b) The changes in the cropland abandoned area and the cropland abandonment rate in KPEYC in the past 20 years.

Fig.3. Distribution of cropland abandonment rate in KPEYC during 2001-2020.

Fig.4. The correlation between the change factors and the cropland abandonment rate in KPEYC.

Fig. 5. Drivers interactive detection results of the mean value in the past 20 years. (a) The interactive detection results of factors. (b) High-value graph of interactive detection.

4. Implications and Generalizability: Discuss the broader implications of your findings, particularly how they might apply to other regions or contexts, to enhance the manuscript's relevance and impact.

Reply:

Thanks for your suggestion. We added implications and generalizability in line 469. The results of the study are first described, the findings of this piece are compared with those of other regions, and their generalisability is discussed. The content is as follows.

modified content:

The research conducted on the issue of CAR in KPEYC reveals that economic factors play a pivotal role in influencing this phenomenon. These economic factors directly impact the reduction in agricultural income and the increase in farmers' consumption levels, thereby affecting their decision to abandon their lands. This finding resonates with the principles outlined in von Thünen's rent theory[25,26], which posits that the marginalization of land due to its poor suitability and diminishing economic returns underlie the decision to abandon land. However, it diverges from the findings of Han Ze[27], who highlighted that cropland abandonment in the karst mountains of Guizhou-Guangxi, China, is predominantly driven by agricultural practices and soil conditions, with soil erosion being a critical determinant of its distribution. The discrepancy could be attributed to the regional variations in the spatial distribution of cropland abandonment or possibly influenced by the different stages of regional economic development; for instance, Yunnan Province has exhibited higher GDP outputs compared to Guizhou Province over the past years.

The methodology employed in this research can indeed be applied across diverse regions globally; however, it is essential to recognize that varying data sets or distinct political cultures may yield different outcomes. For instance, when applied to countries outside of China, discrepancies in land management systems could potentially result in varying primary factors identified through the analysis.

5. Policy Recommendations: Provide detailed policy recommendations based on your findings to demonstrate the practical applications of your research in addressing cropland abandonment.

Reply:

Thanks for your suggestion. We added policy recommendations in line 458. Discussion in terms of economics, rocky desertification management and government policies. The content is as follows.

modified content:

Based on the findings of this study and in alignment with the management policies of the Yunnan Province area in China, the following recommendations are proposed to effectively tackle the issue of cropland abandonment and ensure food security in the region. To address the labor outflow issue induced by economic development, the following measures are suggested that: (i) Rural agricultural subsidies should be enhanced: This would provide much-needed financial support to farmers, especially those cultivating land in mountainous regions. (ii) Preferential treatments should be extended to farmers in marginal lands: For instance, by offering educational and medical benefits to the children of these farmers, thus alleviating the economic burden on their families. (iii) Financial risk mitigation for farmers: By providing low-interest loans and agricultural insurance, farmers in marginal mountainous areas can better withstand financial risks associated with agricultural production. (iv) Expanding sales channels through internet platforms: Utilizing online platforms to open up new sales channels and expand markets for agricultural produce, especially those that are less amenable to long-distance transportation, thereby reducing the sales risk for farmers. (v) Enhanced agricultural technology support: By improving seeds, providing advanced agricultural cultivation technologies, and boosting crop productivity.

For the severely rocky desertified areas in KPEYC, it is imperative to thoroughly implement the concept of an ecological civilization. This includes executing afforestation projects on mountaintops, developing specialized economic industries on the slopes, and undertaking the "slope-to-staircase" project to fully utilize arable land and enhance land productivity. Such measures not only conserve water resources but also reduce soil erosion and tackle pest issues.

While governments at all levels have made preliminary progress in addressing the issue of abandoned cropland, the protection of cropland resources remains a long-term process aimed at promoting the rational use of these resources. To continuously stimulate farmers' enthusiasm for farming, it is necessary to establish a long-term early warning and remediation mechanism for abandoned cropland to prevent the recurrence of this phenomenon.

---

## [Decision Letter · Decision Letter 2]

2 Jul 2024

Spatial-temporal characterization of cropland abandonment and its driving mechanisms in the Karst Plateau in Eastern Yunnan, China, 2001-2020

PONE-D-23-43715R2

Dear Dr. Wang,

We’re pleased to inform you that your manuscript has been judged scientifically suitable for publication and will be formally accepted for publication once it meets all outstanding technical requirements.

Kind regards,

Eda Ustaoglu, PhD

Academic Editor

PLOS ONE

Additional Editor Comments (optional):

Reviewers' comments:

Reviewer's Responses to Questions

**Comments to the Author**

1. If the authors have adequately addressed your comments raised in a previous round of review and you feel that this manuscript is now acceptable for publication, you may indicate that here to bypass the “Comments to the Author” section, enter your conflict of interest statement in the “Confidential to Editor” section, and submit your "Accept" recommendation.

Reviewer #4: All comments have been addressed

2. Is the manuscript technically sound, and do the data support the conclusions?

Reviewer #4: Yes

3. Has the statistical analysis been performed appropriately and rigorously? 

Reviewer #4: Yes

4. Have the authors made all data underlying the findings in their manuscript fully available?

Reviewer #4: Yes

5. Is the manuscript presented in an intelligible fashion and written in standard English?

Reviewer #4: Yes

6. Review Comments to the Author

Reviewer #4: The revised manuscript entitled "Spatial-temporal characterization of cropland abandonment and its driving mechanisms in the Karst Plateau in Eastern Yunnan, China, 2001-2020" looks fine to be accepted for publication.

7. PLOS authors have the option to publish the peer review history of their article (what does this mean?). If published, this will include your full peer review and any attached files.

Reviewer #4: No

---

## [Editor Report · Acceptance letter]

8 Jul 2024

PONE-D-23-43715R2 

PLOS ONE

Dear Dr. Wang, 

I'm pleased to inform you that your manuscript has been deemed suitable for publication in PLOS ONE. Congratulations! Your manuscript is now being handed over to our production team.

Kind regards, 

on behalf of

Dr. Eda Ustaoglu 

Academic Editor

PLOS ONE